# YOLOD: A Target Detection Method for UAV Aerial Imagery

Xudong Luo, Yiquan Wu * and Langyue Zhao

College of Electronic and Information Engineering, Nanjing University of Aeronautics and Astronautics, Nanjing 211106, China; luoxvdong@nuaa.edu.cn (X.L.); zlangyue@nuaa.edu.cn (L.Z.)
* Correspondence: imagestrong@nuaa.edu.cn; Tel.: +86-137-7666-7415

**Abstract:** Target detection based on unmanned aerial vehicle (UAV) images has increasingly become a hot topic with the rapid development of UAVs and related technologies. UAV aerial images often feature a large number of small targets and complex backgrounds due to the UAV's flying height and shooting angle of view. These characteristics make the advanced YOLOv4 detection method lack outstanding performance in UAV aerial images. In light of the aforementioned problems, this study adjusted YOLOv4 to the image's characteristics, making the improved method more suitable for target detection in UAV aerial images. Specifically, according to the characteristics of the activation function, different activation functions were used in the shallow network and the deep network, respectively. The loss for the bounding box regression was computed using the EIOU loss function. Improved Efficient Channel Attention (IECA) modules were added to the backbone. At the neck, the Spatial Pyramid Pooling (SPP) module was replaced with a pyramid pooling module. At the end of the model, Adaptive Spatial Feature Fusion (ASFF) modules were added. In addition, a dataset of forklifts based on UAV aerial imagery was also established. On the PASCAL VOC, VEDAI, and forklift datasets, we ran a series of experiments. The experimental results reveal that the proposed method (YOLO-DRONE, YOLOD) has better detection performance than YOLOv4 for the aforementioned three datasets, with the mean average precision (mAP) being improved by 3.06%, 3.75%, and 1.42%, respectively.

**Keywords:** target detection; UAV aerial imaging; YOLO; attention mechanism; UAV dataset

## 1. Introduction

Small size, light weight, simple operation, energy savings, and low noise are key advantages of an unmanned aerial vehicle (UAV). Its take-off and landing are less restricted by the site, and it can take off and land on playgrounds, roads, or other open ground with good stability and safety. UAVs easily capture images. Due to the different flying heights and viewing angles during shooting, compared with natural images shot at a horizontal angle, aerial photography images of UAVs contain more small targets, and the objects in the images are arranged in a disorderly manner, with random directions and complex backgrounds. UAV systems are easy to transport and have low operating costs. They may carry a variety of sensors for repeated cycle detection, which is convenient for collecting the required data. As deep learning technology has advanced rapidly in recent years, UAV systems have become more intelligent, efficient, and convenient. UAVs have evolved into the ideal equipment for precision agriculture [1–4], smart cities [5–7], and search and rescue [8–10].

Traditional object detection methods employ an exhaustive strategy for region selection [11–13]. Because the target's position, size, and aspect ratio cannot be known, the image is traversed using sliding windows of various scales and aspect ratios. Although all possible target positions are provided, there are issues such as excessive complexity, many redundant windows, and poor area matching that have a significant impact on classification speed and accuracy. Traditional object detection methods use artificially designed features for feature extraction [14,15]. The artificially designed features lack robustness and

adaptability due to the diversity of target morphologies, the uncertainty of illumination changes, and the complexity of target backgrounds. For the above reasons, the detection effect of traditional methods is unstable. They are prone to various constraints, and it is difficult to meet the needs of real-time processing in practical applications.

With the rapid development of deep learning technology in recent years, convolutional neural networks (CNN) have been proven to be more effective in handling a variety of vision tasks. Modern object detection methods can be divided into the following two categories: two-stage detection methods and one-stage detection methods, which have achieved significant improvements in detection accuracy and processing speed. The two-stage detection method consists of two stages, where candidate regions are first proposed through a selective search strategy, and then a classifier built with a convolutional neural network is used to determine the corresponding category score. In the existing two-stage detection models, R-CNN [16] requires a fixed input image size, which restricts the image's aspect ratio. Its training is a multi-stage procedure that is both time and space-consuming, as well as slow in terms of object detection. The selective search process adopted by Fast R-CNN [17] is complex and time-consuming, and does not implement a true end-to-end training mode. Faster R-CNN [18] cannot achieve the effect of the real-time detection of objects. The features of other layers are not fully considered, and the detection performance is obviously insufficient when detecting small objects. The problem of pixel alignment between the network's input and output is ignored, resulting in a general misalignment of the region of interest (ROI) and retrieved features. Furthermore, the number of feature channels after ROI pooling is large, causing the whole connection process to consume a large amount of memory and slow down the model's calculation speed. By cascading, Mask R-CNN [19] and Cascade R-CNN [20] increase detection performance, but the gain is restricted since the information flow across mask branches at various stages is not optimal. R-FCN [21] must build a huge number of score maps, and the network's pace is slow, with a significant computational cost. With the rising requirements for efficiency and real-time performance in the field of target detection, many classical one-stage target detection methods that complete target prediction and localization in one step have been presented. In the existing one-stage detection model, YOLO [22], it is difficult to deal with small targets that appear in groups, the model generalization ability is weak, and the problem of loss function easily leads to obvious positioning errors. SSD [23] adopts a hierarchical structure of pyramid features, which makes it difficult to handle large-scale changes, especially when detecting small-sized objects. DenseBox [24] has difficulty handling overlapping bounding boxes and has low recall. When RetinaNet [25] handles objects of various scales, additional stages are usually required to complete the image classification task. The accuracy of YOLOv2 [26] much improved compared to YOLO, but in subsequent practical applications, the accuracy remained insufficient. The feature extraction network of YOLOv3 [27] lacks fusion operations, and the activation function it uses is not smooth enough, which affects the gradient descent.

Current target detection methods lack outstanding performance on UAV aerial images. Due to the difference in the flying height of the UAV and the shooting angle, the objects in the image contain different texture and shape information than the natural image captured on the ground. Igor Sevo et al. [28] demonstrated the possibility of using CNN for aerial image analysis and proposed a two-stage detection method. The method was tested in image classification tasks. Due to the long detection time, it is necessary to reduce the computation time by configuring multiple GPUs. By combining appearance and motion information, Rodney LaLonde et al. [29] proposed a two-stage spatio-temporal CNN method to improve the target detection effect of UAV-based wide area motion imagery. The frame alignment part introduced in this method to eliminate camera shake will greatly increase the network's computational cost. Yongchao Xu et al. [30] suggested a detection approach to overcome the problem of unpredictable target angles in UAV aerial images, resulting in predicted boxes with more accurate information. However, since the detection model needs to obtain angle information, it requires additional calculation parameters

and calculation time. Danilo Avola et al. [31] presented the MS-Faster R-CNN detection method, which is a multi-stream architecture that efficiently solves the problem of picture quality degradation caused by the UAV's mobility during detection. Its detection speed, however, has to be increased. Jing Zhang et al. [32] analyzed the two factors of real-time processing and detection accuracy, and proposed a UAV aerial image detection method based on a lightweight CNN. To boost detection performance, this approach must optimize the network structure.

In conclusion, new target detection methods have increased in accuracy and speed as a result of the proposal and development of target detection methods. The advantages of two-stage and one-stage detection methods lie in their excellent detection accuracy and speed, respectively. Since the UAV transmits video streams or pictures in real time, the influence of the detection speed needs to be considered, so the YOLO series detection method was selected. The YOLO detection method was improved across several versions, and its detection effect was excellent. Newer YOLO versions are YOLOv4 [33], YOLOv5, and YOLOX [34]. YOLOv5 and YOLOX are larger in scale, have more parameters, and have more complex models than YOLOv4, and they are not separated from the CSPNet structure of the backbone of YOLOv4 and the PANet structure of the neck of YOLOv4. Therefore, this research improved the YOLOv4 target detection method to make it more suitable for target detection in UAV aerial images. We focused on the following four points: (1) because the background of UAV aerial images is cluttered, the backbone's capacity to extract features must be improved; (2) small targets abound in UAV aerial images, which are orientated and grouped in a complex manner. The detection model must be able to emphasize key information while suppressing irrelevant information; (3) the aerial imagery of the UAV has a larger scale due to the UAV's perspective. The detecting model's receptive field must be appropriately expanded; (4) the size of the targets in the UAV aerial images changes as the flying height of the UAV changes. The ability of the feature pyramid for multiscale object detection must be improved. According to the above four requirements, we proposed an improved target detection method suitable for UAV aerial imagery and named it YOLO-DRONE (YOLOD). This work's primary contributions are as follows:

1.  According to the characteristics of the activation function, more appropriate activation functions were used on different layers of the backbone. In shallow and deep networks, HardSwish [35] and Mish [36] activation functions were used, respectively. This choice is beneficial because it reduced model complexity while maintaining good detection accuracy.
2.  We finally chose the EIOU loss [37] function to calculate the loss of the bounding box regression of the model after analyzing and comparing different loss functions. The model's convergence speed quickened as a result, and the positioning effect improved.
3.  The performance of three modules used to enhance the model's receptive field in YOLOv4 was compared, and the spatial pyramid pooling (SPP) module was eventually replaced with the pyramid pooling module [38]. The pyramid pooling module is useful for increasing the model's receptive field and thereby enhancing its detection performance. We also introduced an Adaptive Spatial Feature Fusion (ASFF) module [39] to the model's end to improve multiscale feature fusion, which was useful for detecting objects on different scales.
4.  A new attention module was proposed. The module uses a one-dimensional convolution operation to adaptively determine the number of adjacent channels for each channel, which effectively captures the information interaction between channels. The proposed attention module fully exploits the benefits of global average and global max pooling, allowing the model to extract target features more effectively.
5.  A dataset of forklift trucks based on UAV aerial imagery was established. The dataset consists of 1007 annotated images. This is the first dataset of forklift trucks based on UAV aerial images that we know of.

The following sections make up the remainder of this work. Section 2 gives a brief overview of the previous YOLO series of detection methods and details the proposed

YOLOD improvement strategy. Section 3 introduces the experimental environment, selected datasets, parameter settings, and evaluation metrics and gives detailed experimental procedures and results that verify the superiority of the new method. Section 4 summarizes the new method and indicates research directions for the future.

## 2. Methods

### 2.1. YOLOv3 and YOLOv4 Algorithm Description

The backbone of YOLOv3 is Darknet53. Darknet53 uses a residual structure [40]. First, the width and height of the incoming feature map are compressed by convolution operation with stride $2 \times 2$. Then, two convolution operations with kernel sizes of $1 \times 1$ and $3 \times 3$ are used for feature extraction. Finally, the extracted features are added to the compressed feature map to obtain the output of the residual structure. To improve the model's accuracy, we used the method of deepening the residual network. The residual network is simple to optimize, and the skip connection of the residual block within it alleviates the vanishing gradient caused by increased network depth. Darknet53's convolution block employs the DarknetConv2D structure. Specifically, batch normalization and the LeakyReLU activation function are performed after the convolution operation is completed. Compared with the ReLU activation function, the LeakyReLU activation function [41] adds a non-zero slope at negative values, solving the Dead ReLU problem. At the neck of the model, YOLOv3 extracts multiple feature maps for object detection, which are located in the middle, lower, and bottom layers of the backbone, respectively. After that, the model uses the feature pyramid to perform fusion on the features extracted from different layers, which is conducive to extracting more effective features.

The YOLOv4 detection model is an improved version of YOLOv3 with better detection performance. The backbone of YOLOv4 is CSPDarknet53. Compared with YOLOv3, the improvements in the backbone part are as follows. The Mish activation function replaces the original LeakyReLU activation function in the DarknetConv2D structure. The gradient of the Mish activation function does not vanish and has good smoothness at negative values. The experimental results of D. Misra [36] revealed that the detection accuracy using the Mish activation function was better than the Swish and ReLU activation functions in the same network. The backbone uses the CSPNet structure [42], which splits the original stack of residual blocks into two parts. One part is used for the superposition of the residual structure, and the other part is added to the previous part after a small number of convolution operations. The CSPNet structure enhances the learning ability of the network, can keep the network lightweight while maintaining good accuracy, and reduces computational bottlenecks and memory costs. At the model neck, YOLOv4 uses the SPP module and PANet structure [43]. After the output of CSPdarknet53's last feature map, the SPP module is added. This module processes data using four different max pooling scales, which helps to expand the receptive field and distinguish important contextual features. The PANet adds a bottom-up feature fusion step, which overcomes the limitations of classic FPN networks due to unidirectional information flow. This structure increases information usage, enhances the efficiency with which information is disseminated, and is useful for recurring feature extraction. In the feature utilization part, YOLOv4 behaves the same as YOLOv3, extracting three feature maps located in the middle, lower, and bottom layers of the backbone. The output feature maps have the shapes (52, 52, 256), (26, 26, 512), and (13, 13, 1024) when the input size is $416 \times 416$ pixels. Since YOLOv4 has three prior bounding boxes for each feature point, the shapes of the feature maps output by YOLO Head are (52, 52, $3 \times (4 + 1 +$ number of classifications)), (26, 26, $3 \times (4 + 1 +$ number of classes)) and (13, 13, $3 \times (4 + 1 +$ number of classes)). The number "4" represents the priori boxes' adjustment parameters, "1" indicates if the priori boxes contain objects, and "number of classes" indicates the detection score of each category. Mosaic data augmentation is utilized during YOLOv4 training to enrich the background of recognized objects and boost batch normalization efficiency. Label smoothing is used to prevent model overfitting.

Using the CIOU loss function makes the target regression more stable. Figure 1 shows YOLOv4's architecture.

**Figure 1.** The YOLOv4's architecture.

### 2.2. Algorithm Design and Improvement

The aerial image of the UAV includes more small targets than the natural image acquired from the horizontal perspective due to the UAV's flying height and shooting angle. In addition, the objects in the UAV aerial images were arranged in a disorderly manner, with random directions and complex backgrounds. The shortcomings of the present detection methods were described in the preceding section, and they still have a lot of room for improvement. Based on the properties of UAV aerial images, our method aimed to achieve the following goals: (1) enhance the feature extraction capability of the backbone; (2) emphasize important information in UAV aerial images and suppress irrelevant information; (3) increase the receptive field of the detection model; and (4) enhance the target detection ability of the feature pyramid for multi-scale targets. Our improvements mainly concerned the backbone and neck of the model.

#### 2.2.1. Improvement of Detection Model Backbone

Our backbone improvements to YOLOv4 were as follows: (1) choosing more suitable activation functions; (2) applying a new loss function; and (3) adding new attention modules.
(1) Choosing more suitable activation functions.

The activation function was utilized in the detection model to raise the nonlinear factors so that the convolutional neural network was able to approximate any nonlinear function. This allows the model to fully utilize the advantages of multi-layer superposition and improve its expressive capacity. The following two issues must be considered while selecting an activation function: (1) whether it improves gradient propagation; and (2) the cost of function calculation.

Early activation functions include Sigmoid and Tanh activation functions. The functional formula and derivation of the Sigmoid activation function are shown in Equations (1) and (2), and the functional formula and derivation of the Tanh activation function are shown in Equations (3) and (4):

$$\text{Sigmoid} = f(x) = \frac{1}{1 + e^{-x}} \tag{1}$$

$$f'(x) = \frac{1}{1 + e^{-x}} \times \left(1 - \frac{1}{1 + e^{-x}}\right) = f(x) \times (1 - f(x)) \tag{2}$$

$$\text{Tanh} = f(x) = \frac{e^x - e^{-x}}{e^x + e^{-x}} \tag{3}$$

$$f'(x) = 1 - \frac{(e^x - e^{-x})^2}{(e^x + e^{-x})^2} = 1 - f^2(x) \tag{4}$$

The Sigmoid and Tanh activation functions are smooth and differentiable, and the derivative calculation is convenient. The output of the sigmoid activation function is bounded between 0 and 1, which makes it stable for some larger inputs as well. The output of the Tanh activation function is stable between $[-1, 1]$, symmetric about the 0 center, and the gradient is larger near 0, which is beneficial to distinguish small feature differences. The outputs of the Sigmoid activation function are all positive values, which causes zigzag shaking during gradient descent and in turn reduces the gradient descent speed. The output is not centered at 0, reducing weight update efficiency. Both Sigmoid and Tanh activation functions have nonlinear saturation regions, which easily cause the phenomenon of vanishing gradient during backpropagation. Since the derivative of Sigmoid has a maximum value of 0.25 and the derivative of Tanh has a maximum value of 1, the vanishing gradient of Tanh is slightly smaller than that of Sigmoid.

Since its inception, the ReLU activation function [40] has become one of the most widely utilized activation functions. The functional formula and derivation of the ReLU activation function are shown in Equations (5) and (6):

$$\text{ReLU} = f(x) = \max(0, x) \tag{5}$$

$$f'(x) = \begin{cases} 1 & , \text{ if } x \geq 0 \\ 0 & , \text{ if } x < 0 \end{cases} \tag{6}$$

Because the ReLU activation function only requires information on whether the input is greater than 0, the calculation is simple and quick. This function converges much faster than the Sigmoid and Tanh functions. The ReLU activation function solves the gradient dispersion problem in the positive interval; however, in the process of back propagation, if the input is negative, the gradient is zero, and there is a Dead ReLU problem. This will cause some units to remain inactivated and the corresponding parameters to never be updated. For the Dead ReLU problem, PReLU [44], Elu [45], Leaky ReLU [41], and other improved methods based on this function were proposed. The functional formulations of the activation functions of PReLU, ELU, and Leaky ReLU are shown in Equations (7)–(9):

$$\text{PReLU} = \begin{cases} y_i & , \text{ if } y_i \geq 0 \\ a_i y_i & , \text{ if } y_i < 0 \end{cases} \tag{7}$$

$$\text{Elu} = \begin{cases} x & , \text{ if } x \geq 0 \\ a(e^x - 1) & , \text{ if } x < 0 \end{cases} \tag{8}$$

$$\text{Leaky Relu} = \begin{cases} x & , \text{if } x \geq 0 \\ ax & , \text{if } x < 0 \end{cases} \tag{9}$$

The parameter ai in PReLU is usually limited between 0 and 1. If $a_i = 0$, it is converted to a ReLU function; if $a_i > 0$, it is converted to a Leaky ReLU function; if $a_i$ is a learnable parameter, it is expressed as a PReLU function. The PReLU, ELU, and Leaky ReLU activation functions all have slopes in the negative domain, thus avoiding the Dead ReLU problem. PReLU requires additional computation and increases the risk of overfitting due to the introduction of additional hyperparameters. Elu involves exponential operations, so the calculation is complicated and slow. Leaky ReLU lacks robustness to noise.

The Swish activation function [46] is an activation function proposed by Google in 2017. The functional formula and derivation of this activation function are shown in Equations (10) and (11):

$$\text{Swish} = f(x) = x \cdot \text{Sigmoid}(\beta x) \tag{10}$$

$$f'(x) = x \cdot \text{Sigmoid}(x) + \text{Sigmoid}(x)(1 - x \cdot \text{Sigmoid}(x)) = f(x) + \text{Sigmoid}(x)(1 - f(x)) \tag{11}$$

The functional formula of the Swish shows that when $\beta = 0$, Swish = 0.5$x$, and when it tends to infinity, Swish is transformed to ReLU, indicating that this function is akin to a smooth function between a linear function and ReLU. Swish is smooth and non-monotonic; the output has lower bounds and no upper bounds; and the effect on deep networks is better than the ReLU activation function. However, its calculation speed is slightly slower than ReLU's.

The HardSwish activation function [35] is proposed in MobileNetV3. The functional formulation and derivation of this activation function are shown in Equations (12) and (13):

$$\text{HardSwish} = f(x) = \begin{cases} 0 & , \text{if } x \leq -3 \\ x & , \text{if } x \geq 3 \\ \frac{x(x+3)}{6} & , \text{otherwise} \end{cases} = x\frac{\text{ReLU6}(x+3)}{6} \tag{12}$$

$$f'(x) = \frac{\text{ReLU6}(x+3)}{6} + \frac{x}{6} \cdot \text{ReLU6}'(x+3) \tag{13}$$

Compared with Swish, the HardSwish activation function has better stability and a faster calculation speed.

Mish [36] is the activation function in the YOLOv4 backbone. The functional formulation and derivation of this activation function are shown in Equations (14) and (15):

$$\text{Mish} = f(x) = x \cdot \text{Tanh}(\ln(1+e^x)) = x \cdot \frac{(1+e^x)^2 - 1}{(1+e^x)^2 + 1} = x \cdot \frac{y^2 - 1}{y^2 + 1} \tag{14}$$

$$f'(x) = \frac{4y(y-1)}{(y^2+1)^2} \cdot x + \frac{y^2 - 1}{y^2 + 1} \tag{15}$$

The Mish activation function improves training stability, average accuracy, and peak accuracy significantly. However, this function is computationally difficult.

In deep networks, especially after layer 16, the accuracy of the ReLU activation function will drop rapidly. The same problem occurs with the Swish activation function after 21 layers. However, the Mish activation function still maintains good accuracy. This small gap is magnified after passing through the deep network, showing a large performance improvement. Figure 2 depicts an intuitive comparison of the mathematical models of the Leaky ReLU, Swish, HardSwish, and Mish activation functions, respectively.

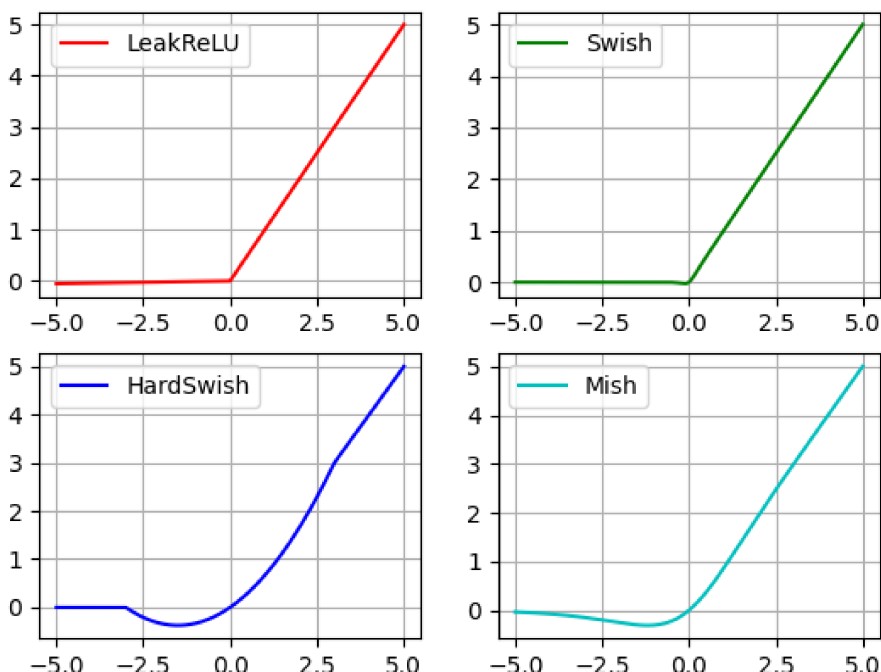

**Figure 2.** Mathematical models of Leaky ReLU, Swish, HardSwish, and Mish activation functions are compared intuitively.

In summary, combined with the characteristics of each activation function, we found that the HardSwish activation function has good detection accuracy and computational complexity in shallow networks, while the Mish activation function has a greater contribution in deep networks. Under the trade-off between model complexity and detection accuracy, the activation function in the original YOLOv4 backbone was modified. The activation functions of HardSwish and Mish were used in the first two layers and the last four layers of the backbone, respectively.

(2) Applying a new loss function.

In the YOLOv4 detection method, bounding box regression (BBR) is used to locate objects. The choice of loss function is critical because it can seriously affect the performance of target localization. The early use of the L2 norm loss to compute the loss for bounding box regression [24,47], as shown in Equation (16), represents the sum of the squares of the differences between the predicted and ground-truth values. Jiahui Yu et al. [48] pointed out that the L2 norm loss has two main flaws. First, the correlation between the coordinates of the bounding box is torn, resulting in the inability to obtain an accurate bounding box. Second, the loss is not normalized, so there is an imbalance between objects of different sizes. This causes the model to focus more on large objects while ignoring small objects.

$$L_{l2} = \sum_{i=1}^{N} (y_i - f(x_i))^2 \qquad (16)$$

$y_i$ is the genuine value, $f(x_i)$ is the forecast value, and $L_{l2}$ is the loss value in the equation above.

In order to solve these defects, in subsequent research, a loss based on intersection over union (IOU) was proposed. There are mainly IOU loss [48], GIOU loss [49], DIOU loss [50], CIOU loss [50] and EIOU loss [37]. Scale invariance, satisfying non-negativity, identity, symmetry, and triangular inequality are all characteristics of IOU, which is a regularly used indicator in object detection. However, if the two bounding boxes do not intersect, the IOU loss cannot accurately reflect the distance between them. Hence, the IOU loss cannot accurately describe the degree of overlap. On the basis of IOU characteristics, GIOU introduces the minimum circumscribed rectangle, which overcomes the problem

of the loss value being 0 when the bounding boxes do not overlap. However, when the bounding boxes are contained, GIOU degenerates into IOU. When the bounding boxes are in a state of intersection, the GIOU loss converges slowly in both the horizontal and vertical dimensions. To solve GIOU's slow convergence speed, DIOU directly returns the straight-line distance between the center points of the bounding boxes, which accelerates the convergence speed. However, the bounding boxes' aspect ratio is not taken into account in the regression procedure, and the DIOU loss still needs to be improved in terms of accuracy. The CIOU loss adds the loss of length and width to the DIOU loss, which makes the predicted boxes more consistent with the real boxes. However, the aspect ratio in CIOU is a relative value, which is ambiguous. The penalty term of CIOU only reflects the difference in length and width, which may optimize the similarity in an unreasonable way. The EIOU loss is improved following the CIOU loss. EIOU and EIOU loss are calculated using Equations (17) and (18):

$$EIOU = IOU - \frac{\rho^2\left(b, b^{gt}\right)}{c^2} - \frac{\rho^2\left(w, w^{gt}\right)}{C_w^2} - \frac{\rho^2\left(h, h^{gt}\right)}{C_h^2} \tag{17}$$

$$L_{EIOU} = 1 - EIOU \tag{18}$$

The width and height of the minimum circumscribed rectangle covering the ground-truth box and the prediction box are represented by CW and CH in the preceding formula. EIOU loss is divided into the following three parts: IOU loss, distance loss, and location loss. EIOU directly reduces the difference between the width and height of the bounding boxes, which increases the speed of convergence and improves the position effect. YOLOv4 uses CIOU loss when computing bounding box regression. We then replaced it with an EIOU loss.

(3) Adding new attention modules.

The performance of deep convolutional neural networks has been proven to benefit from attention mechanisms. Representative attention modules include the Squeeze-and-Excitation (SE) module [51], the Convolutional Block Attention Module (CBAM) [52], and the Efficient Channel Attention (ECA) module [53], etc.

The SE module improves the feature quality of the network through the interdependence between channels. Through this module, the network can selectively emphasize informative features and suppress irrelevant features. This module is generic and has different effects at different depths throughout the network. Features are excited in a class-independent manner in shallow networks. In deep networks, the module becomes increasingly specialized, responding to different inputs in a highly class-specific manner. The CBAM module computes the attention map along two different dimensions and sequentially, and then multiplies the attention map with the input feature map to obtain the refined features. The ECA module omits the fully connected layer in the SE module and directly performs global average pooling by channels. An adaptive method was then used to determine the number of adjacent channels for each channel, which was proportional to the signal dimension. This method ensured that the module was both efficient and effective.

In the process of determining the weight of each channel, the SE module uses two fully connected layers, which reduces the complexity of the model. However, this procedure seems to be of limited help in capturing the interactions between all channels and thus may be redundant. The channel attention sub-module in the CBAM module adds global maximum pooling on the basis of global average pooling in the past, which enriches the features of the target. However, the CBAM module increases the computational complexity of the module due to the existence of two connected sub-modules. The ECA module avoids the operation of dimensionality reduction and provides an efficient adaptive method to obtain the number of adjacent channels. However, the ECA module only performs the global average pooling operation, ignoring the gain brought by the global maximum pooling.

We proposed the Improved Efficient Channel Attention (IECA) module as a new attention module, as shown in Figure 3. The IECA attention module first utilizes global average pooling and global max pooling operations to obtain channel information for feature maps. Then, the one-dimensional convolution operation is used to adaptively determine the number of adjacent channels for each channel, and the calculated results are added to obtain the attention map of the feature map. Then, the weight of each channel is calculated using the Sigmoid function. Lastly, the final result is generated by multiplying the obtained weights by the input feature map.

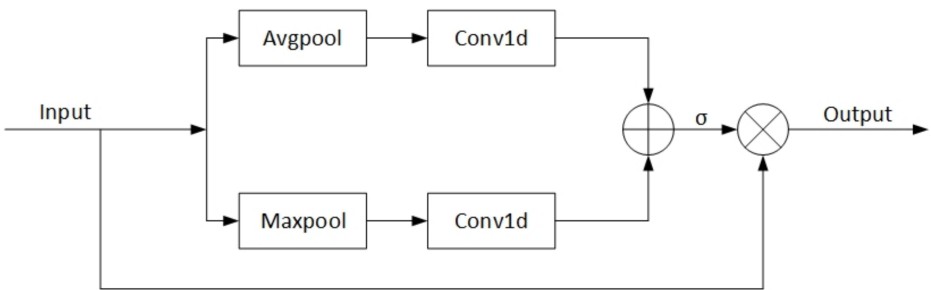

**Figure 3.** The IECA Module's architecture.

2.2.2. Improvement of the Detection Model Neck

Our neck improvements to YOLOv4 were as follows: (1) the pyramid pooling module [38] was used to replace the SPP module in YOLOv4; and (2) an Adaptive Spatial Feature Fusion (ASFF) module [39] was added at the end of PANet.

(1) Use the pyramid pooling module to replace the SPP module in YOLOv4.

The SPP module used in the YOLOv4 detection model uses pooling kernels of different sizes to perform a max-pooling operation and then concatenates the individual results. The pooling kernel sizes used are 1, 5, 9, and 13, respectively. Originally, Kaiming He et al. [54] proposed the SPP module in order to solve the limitation that the CNN must input pictures of a specified size, which avoids the problem of information loss caused by image clipping. Because the proposed SPP module's output is a one-dimensional matrix, it is unsuitable for the Fully Convolutional Network (FCN), so Joseph Redmon and Ali Farhadi revised it. It was modified as a cascade of max-pooling outputs of different pooling kernels. This structure helps in the expansion of the receptive field and the separation of important contextual features.

The Receptive Field Block (RFB) module [55] and the pyramid pooling module, among others, are employed to increase the model's receptive field. Songtao Liu et al. proposed the RFB module, which was inspired by the receptive field organization in the human visual system. Multiple convolution branches make up this module, with different-sized convolution kernels providing different-sized receptive fields and dilated convolution providing individual eccentricities for each receptive field. The output of all branches is cascaded at the end, and the final result is obtained by adjusting the number of channels through a convolution operation. The dilated convolutions used in the RFB module are beneficial for expanding the receptive field and capturing multi-scale contextual information. However, they affect the continuity of information and cause local information loss. In addition, since dilated convolutions sparsely sample the input signal, there is a lack of correlation between the information obtained by long-distance convolution. The Pyramid Scene Parsing Network (PSPNet) uses the pyramid pooling module to generate feature maps of various sizes. This module contains operations for average pooling with various strides and output sizes. Bilinear interpolation is utilized to adapt each feature map to the same size during the operation, and then all feature maps are stacked. The pyramid pooling module comprises four different scales of features, which are separated into $1 \times 1$, $2 \times 2$, $3 \times 3$ and $6 \times 6$ sub-regions in sequence from coarse to fine. This helps to fully utilize each region's contextual information, resulting in more accurate prediction results. Figure 4 shows the pyramid pool module's architecture.

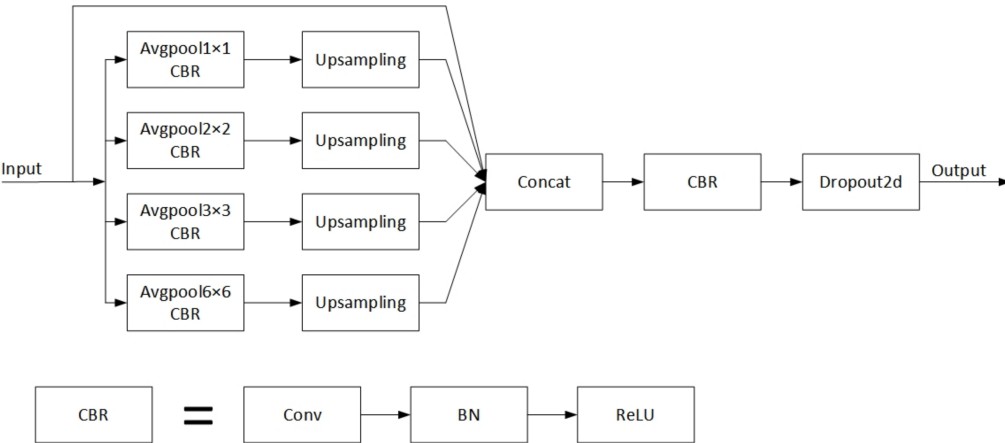

**Figure 4.** The pyramid pooling module's architecture.

In our experiments, we found that replacing the SPP module in YOLOv4 with the pyramid pooling module improved the distinguishing and robustness of features and obtained more reliable prediction results.

(2) An Adaptive Spatial Feature Fusion (ASFF) module at the end of PANet was added.

Pyramid-shaped feature representations are often used to address the challenges posed by scale changes in object detection. However, when using feature pyramids to detect objects, inconsistencies between different feature scales can cause unnecessary conflicts. For example, YOLOv4 detects small and large objects using feature maps of sizes $52 \times 52$ and $13 \times 13$, respectively, when the input image resolution is $416 \times 416$. If an image contains objects of different sizes at the same time, the small object will be regarded as the target on the $52 \times 52$ feature map, but it will be regarded as the background on the $13 \times 13$ feature map. The same is true for large objects. The ASFF module is used to address the inconsistency of feature pyramid characteristics in one-stage detectors. This module retains useful information from each feature map and combines them by filtering feature maps of different sizes. The ASFF module first resizes all feature maps to the same size, and then finds the optimal feature fusion method during model training. At each spatial location, feature maps of different sizes will be adaptively fused to filter out the contradictory information carried by the location and to retain those more discriminative clues. As shown in Figure 5, we added this module at the end of the neck of the YOLOv4 model, which promoted feature extraction and increased the model's detection accuracy.

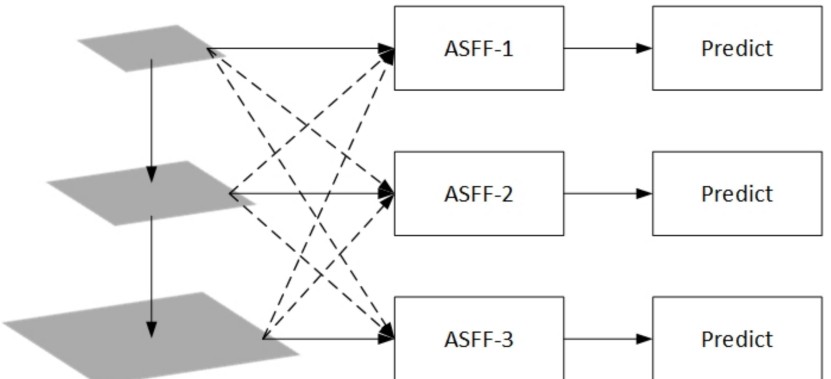

**Figure 5.** Add ASFF module at the end of the neck.

In summary, we made some improvements on the basis of the YOLOv4 model, replaced some modules in the original model, and added some new modules. The architecture of YOLOD is shown in Figure 6. When the input UAV aerial imagery is $416 \times 416$ pixels in size, it first passes through the backbone to produce three feature maps with sizes of

$52 \times 52$, $26 \times 26$, and $13 \times 13$, respectively. Then, the feature maps extracted by the backbone undergo feature fusion through the neck to deepen the feature extraction. Finally, the model's head outputs the final forecast result.

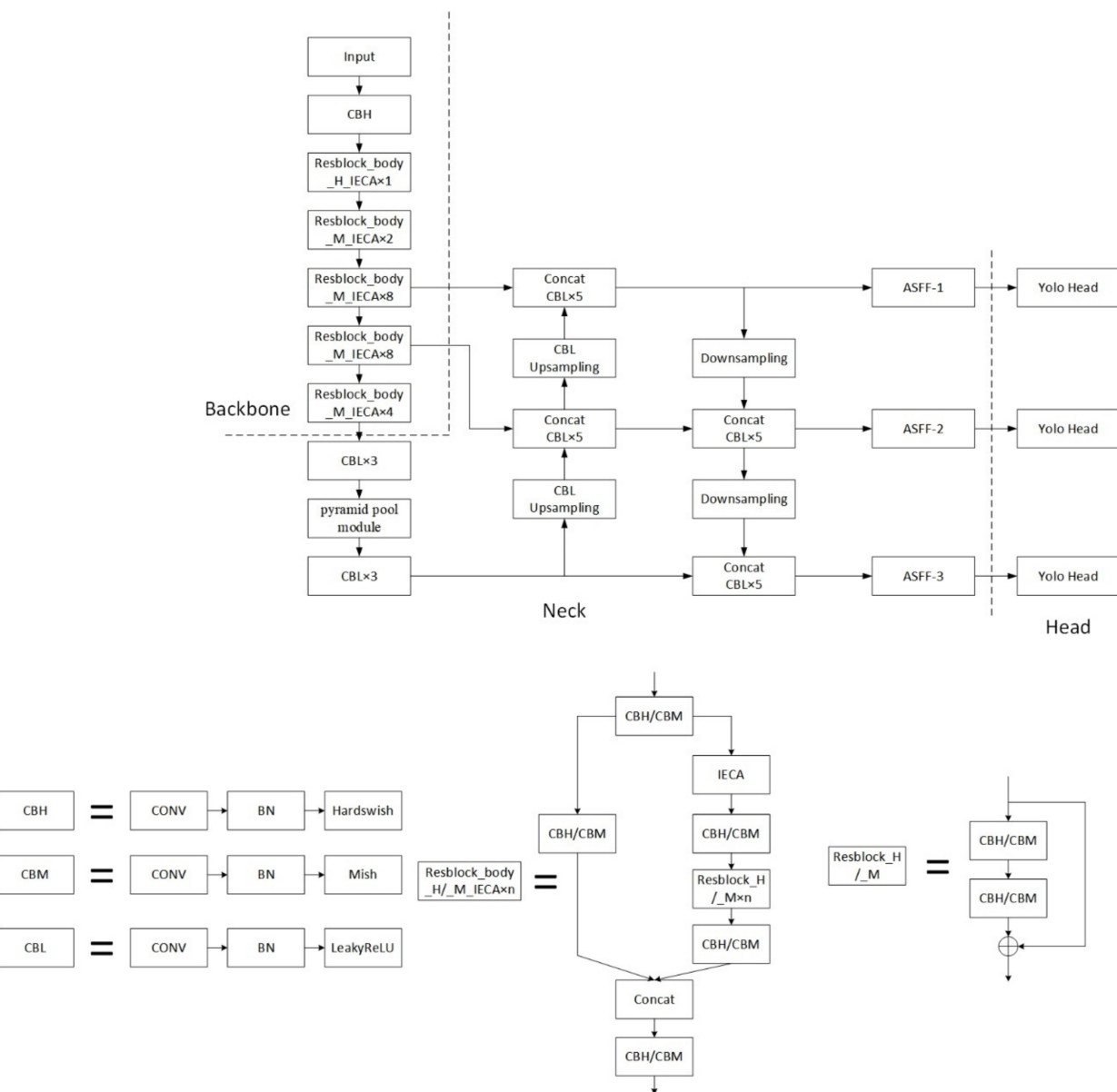

**Figure 6.** The YOLOD's architecture.

## 3. Experiments and Results

In this part, we ran a number of experiments on both the generic dataset and the UAV aerial picture dataset. The selected datasets were PASCAL VOC, VEDAI [56], and the forklift datasets. Ablation experiments were used to verify the effectiveness of our proposed attention module and the improved portions. The proposed method was then compared to a number of different state-of-the-art detectors to verify that it is superior. Finally, experiments on the UAV aerial image dataset were conducted to verify the effectiveness of our proposed method in UAV aerial imaging. Figure 7 shows some examples of the VEDAI dataset and the forklift dataset.

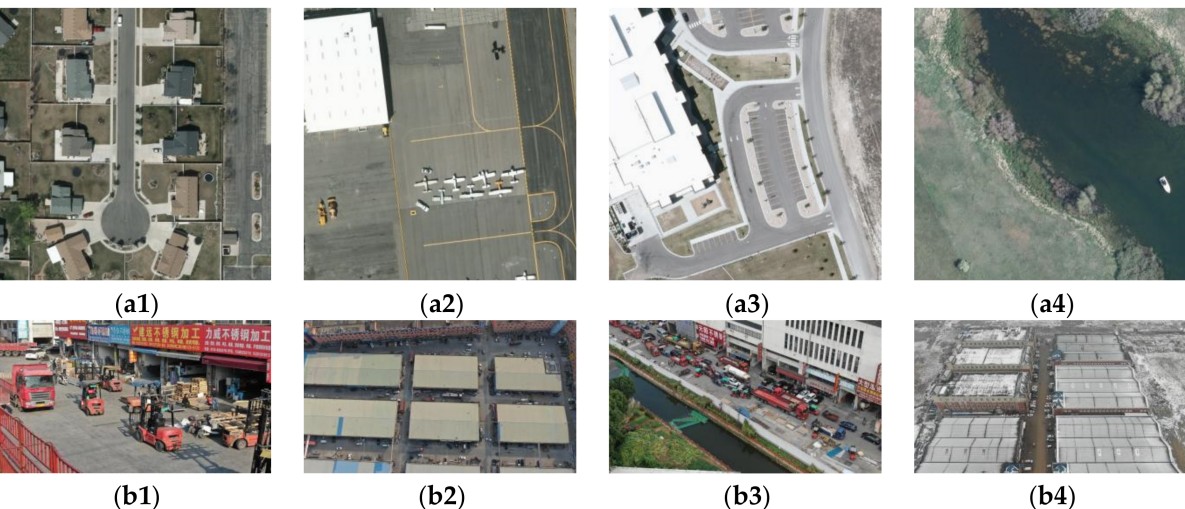

**Figure 7.** Some examples from the datasets: (**a1–a4**) and (**b1–b4**) are pictures from the VEDAI and forklift datasets, respectively.

### 3.1. Experimental Environment and Training Parameter Settings

The frameworks we used were Python 3.8.12, torch 1.8.1, and torchvision 0.9.1. The operating system was Windows 10. The CPU was an i7-10700k. The graphics card was an NVIDIA GeForce RTX 2070 Super. For each set of experiments, we used some of the same initial training parameters as shown in Table 1. Among them, the optimizer chooses SGD. The SGD optimizer is fast to train with low memory overhead. In addition, after adding momentum, the convergence rate is faster and more stable than before, and it helps to reduce the oscillation at the saddle point. Ashia C. Wilson's [57] experimental results show that in the non-adaptive method, SGD finds solutions in CNN training that are more general and have smaller test errors than adaptive methods such as AdaGrad, RMSProph, and Adam.

**Table 1.** Some initial training parameters.

| Input Size | Optimizer | Momentum | Learning Rate Decay | Batch Size | Training Epoch | Training, Validation, and Test Set Ratio |
|---|---|---|---|---|---|---|
| 416 × 416 | SGD | 0.937 | 0.0005 | 4 | 100 | 8:1:1 |

### 3.2. Dataset

For ablation experiments and comparison experiments with other advanced detection methods, we used the Pascal VOC dataset. The Pascal VOC dataset is divided into 20 common categories and includes the VOC2007 and VOC2012 general object detection datasets, which consist of 4952 and 16,552 annotated images, respectively.

The UAV aerial image dataset we used was the VEDAI dataset. This dataset, proposed by Sebastien Razakarivony et al. consists of four distinct subsets. We selected small-sized color images to train the model. The selected image size was 512 × 512 and consisted of 1246 annotated images. The dataset was divided into nine categories, including "other". The images included a variety of different backgrounds, and the detected vehicles had different angles.

We built a forklift dataset based on drone aerial imagery. The images in the dataset were taken by two professional UAV pilots who hold civilian UAV pilot certificates. The UAV used for the shooting is a DJI Mavic 2. The shooting location is Dongfang Steel City in Xishan District, Wuxi City. The target of the dataset was a forklift truck and consisted of 1007 annotated images. The image includes a forklift taken from various perspectives and altitudes by the UAV. Forklifts are mainly small targets with different occlusions, rotations, and dense arrangements. In addition, the datasets had different backgrounds, including

different shooting times, shooting locations, and weather at the time of shooting. Since our experiments relied on the accurate labeling of each object in the forklift dataset, we invited two drivers responsible for filming to participate in the labeling of the dataset. First, we annotated the entire dataset. To ensure consistency, the annotations were checked separately by two drivers, who corrected errors such as under-, missing-, or mislabeled annotations.

*3.3. Evaluation Indicators*

Table 2 shows the parameters used to quantify the correctness of each model output in object detection. Table 3 shows the model detection performance evaluation indicators. In the experiments in this paper, we focused on mAP as a measure of model performance.

$$P = Precision = \frac{TP}{TP + FP} \tag{19}$$

$$R = Recall = \frac{TP}{TP + FN} \tag{20}$$

$$F1 = \frac{2 \times Recall \times Precision}{Recall + Precision} \tag{21}$$

$$AP = \int P(R) \, \mathrm{d}R \tag{22}$$

$$mAP = \frac{1}{C} \sum_{j}^{C} AP_j \tag{23}$$

**Table 2.** Parameters used to quantify the correctness of the model output.

| Parameter | Actual | Predicted |
|---|---|---|
| TP (True Positive) | Positive | Positive |
| FP (False Positive) | Negative | Positive |
| FN (False Negative) | Positive | Negative |
| TN (True Negative) | Negative | Negative |

**Table 3.** The evaluation indicators of the model detection performance.

| Evaluation Indicators | Description | Significance | Calculation |
|---|---|---|---|
| Precision (P) | Ratio between predicted positive samples and actual positive samples. | The probability that all predicted positive samples are actually positive samples. | Equation (19) |
| Recall (R) | Ratio of positive samples with accurate predictions to all positive samples with accurate predictions. | The ability of the classifier to identify positive samples. | Equation (20) |
| F1 score | Harmonic mean of precision and recall. | F1 is a compromise between precision and recall. | Equation (21) |
| Average Precision (AP) | Average of the precision of a certain category. | Indicates how well the model recognizes a certain category. | Equation (22) |
| Mean Average Precision (mAP) | Average of the sum of the APs of all categories in the data set. | Measure how well the model is on average across all categories. | Equation (23) |

*3.4. Experimental Results*

3.4.1. Ablation Experiments

With ablation experiments, we checked the usefulness of the new attention module and the rest of the model's improvements. Ablation experiments were performed on the backbone and neck of the model, respectively, because our model improvements primarily focused on them. The experiments in this part were conducted on the VOC2007 dataset.

The training was carried out using the YOLOv4 backbone's pre-training weights, and the model's backbone was not frozen, so the unfreezing training was performed directly.

We made the following improvements to the backbone: In the first step, appropriate activation functions were selected and a new loss function was applied, then they were merged together. In the second step, the ECA attention module and our proposed IECA module were added, respectively. The modifications we made to the neck are as follows: First, we added the ASFF module at the end of PANet. Second, we replaced the original SPP module with the pyramid pool module. Table 4 shows the experimental results of the ablation experiments.

**Table 4.** Results of ablation experiments.

| Method | Modify Activation Functions | Modify Loss Function | Modify Activation Functions and Loss Function | Add ECA Attention Module | Add our Proposed IECA Module | Add ASFF Module | Replacing the SPP Module with the Pyramid Pool Module | mAP |
|---|---|---|---|---|---|---|---|---|
| YOLOv4 | | | | | | | | 80.13% |
| | √ | | | | | | | 80.79% |
| | | √ | | | | | | 81.09% |
| | | | √ | | | | | 81.75% |
| | | | √ | √ | | | | 82.07% |
| | | | √ | | √ | | | 82.31% |
| | | | | | | √ | | 82.56% |
| | | | | | | √ | √ | 82.81% |
| YOLOD | | | √ | | √ | √ | √ | 83.19% |

From Table 4, we can find that the mAP of YOLOv4 is 80.13%, the mAP of our proposed method is 83.19%, and the mAP improved by 3.06%. Specifically, in the modification of the backbone part, when suitable activation functions are used, the mAP of the model improves to 80.79%. When the new loss function is applied, the mAP of the model improves to 81.09%. When these two parts are merged together, the mAP of the model improves to 81.75%. Based on this step, we added the ECA attention module and our proposed IECA module, and the detection accuracy was improved to 82.07% and 82.37%, respectively. Through comparison, it can be found that our improved attention module is more helpful for the improvement of model detection accuracy. In the modification of the model neck, the mAP was increased to 82.81% through two-part improvements. Finally, we combined all of the backbone and neck improvements to obtain the final detection model. The mAP of this model improved to 83.19%.

In ablation experiments, we visualized YOLOv4, the model after adding ECA, and the model after adding IECA. Using the results visualized using Grad-CAM [58], it was clear that the network uses the extracted features to make judgments. The detection accuracy and Grad-CAM masks for several classes in the dataset are given in Figure 8. After adding the IECA module, the model's Grad-CAM mask can better cover the target area, and the extracted features are more discriminative. In addition, it can be seen from the numerical value of the P that the detection accuracy also improved.

### 3.4.2. Comparison with Other Object Detection Algorithms

We verified the superiority of the proposed method by comparing it with several other advanced detection methods. The detection methods chosen for comparison were: Faster R-CNN, SSD, YOLOv3, YOLOv4-Tiny, YOLOv4, YOLOv5l, and YOLOv5x. The experiments in this part were conducted on the VOC2007 dataset. When training each detection model, the pre-training weights of each model backbone were used, and training was performed on this basis. We did not freeze the backbone of the model and proceeded directly to unfreeze training. Table 5 shows the experimental results of the comparative experiments.

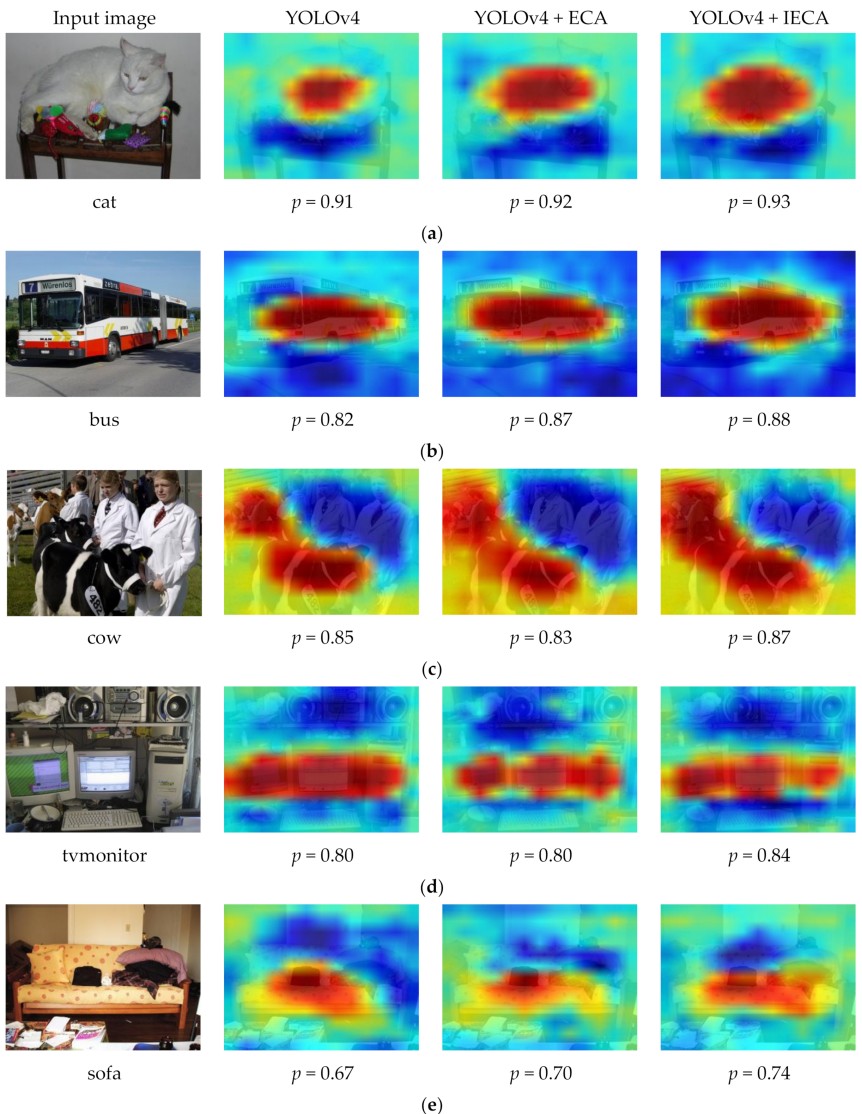

**Figure 8.** Results after visualizing the model: (**a**–**e**) show the detection accuracy and corresponding masks for different classes.

**Table 5.** Experimental results of comparative experiments.

| Method | Backbone | mAP |
| --- | --- | --- |
| Faster R-CNN | ResNet50 | 44.39% |
| SSD | VGG16 | 72.26% |
| YOLOv3 | Darknet53 | 69.00% |
| YOLOv4-Tiny | Tiny CSPDarknet53 | 50.43% |
| YOLOv4 | CSPDarknet53 | 80.13% |
| YOLOv5l | CSPDarknet_l | 80.01% |
| YOLOv5x | CSPDarknet_x | 82.87% |
| YOLOD | Figure 6 | 83.19% |

Table 5 shows that our proposed detection model outperformed many existing advanced object detection approaches in terms of mAP.

3.4.3. Experimental Results on the UAV Aerial Image Dataset

We improved on the YOLOv4 model and proposed a new UAV aerial image target detection method. As a result, the performance of YOLOv4 and the proposed method on UAV aerial images was compared in this experiments. Inspired by the idea of transfer

learning, we first trained YOLOv4 and our proposed model on the VOC2007 and VOC2012 datasets. The current stage of training was based on the YOLOv4 backbone's pre-trained weights. The VOC2007 and VOC2012 datasets have a total of 21,504 annotated images. The second step is to use the weights trained on the VOC2007 and VOC2012 datasets as the pre-training weights of the model and train them on the VEDAI dataset and the forklift dataset, respectively. We then set the epoch to 500.

Table 6 shows the experimental results for the VOC2007 and VOC2012 datasets. Our proposed method improved the mAP by 1.99%.

**Table 6.** Results of training on the VOC2007 + VOC2012 dataset.

| Method | Dataset Used for Training | mAP |
|--------|---------------------------|-----|
| YOLOv4 | VOC2007 + VOC2012 dataset | 87.35% |
| YOLOD | VOC2007 + VOC2012 dataset | 89.34% |

Tables 7 and 8 show the results for the UAV aerial image dataset. Table 7 shows the VEDAI dataset detection results, and Table 8 shows the forklift dataset detection results.

**Table 7.** Results of training on the VEDAI dataset.

| Method | Plane | Camping Car | Car | Boat | Truck | Van | Pickup | Tractor | mAP |
|--------|-------|-------------|-----|------|-------|-----|--------|---------|-----|
| YOLOv4 | 0.80 | 0.65 | 0.59 | 0.54 | 0.32 | 0.30 | 0.26 | 0.17 | 45.73% |
| YOLOD | 0.84 | 0.61 | 0.57 | 0.51 | 0.44 | 0.41 | 0.32 | 0.23 | 49.12% |

**Table 8.** Results of training on the forklift dataset.

| Method | mAP |
|--------|-----|
| YOLOv4 | 70.55% |
| YOLOD | 71.97% |

Tables 7 and 8 show that our proposed methods had a higher mAP. Among them, on the VEDAI dataset, the mAP of YOLOv4 is 45.37%, and the mAP of our proposed method is 49.12%, demonstrating an improvement of 3.75%. The mAP of YOLOv4 is 70.55% on the forklift dataset, and the mAP of our proposed method is 71.97%, an improvement of 1.42%. In Figures 9 and 10, we show some post-detection results.

It can be seen from the above experiments that the proposed method led to the highest improvement on the VEDAI UAV aerial image dataset, which was 3.75%. The second was the improvement in the PASCAL VOC dataset, which was 3.06%. On the forklift dataset, the improvement was the least at 1.42%. The reasons for the analysis are as follows: (1) since the design of the YOLOD detection method is mainly aimed at the characteristics of the target in the UAV aerial image, the mAP improvement is the highest; (2) the PASCAL VOC dataset contains far more images than the VEDAI and forklift datasets, which makes the improvement on this dataset second only to VEDAI; (3) at present, the forklift dataset we established contains few pictures and needs to be supplemented later. The detection performance will also improve as the number of images increases.

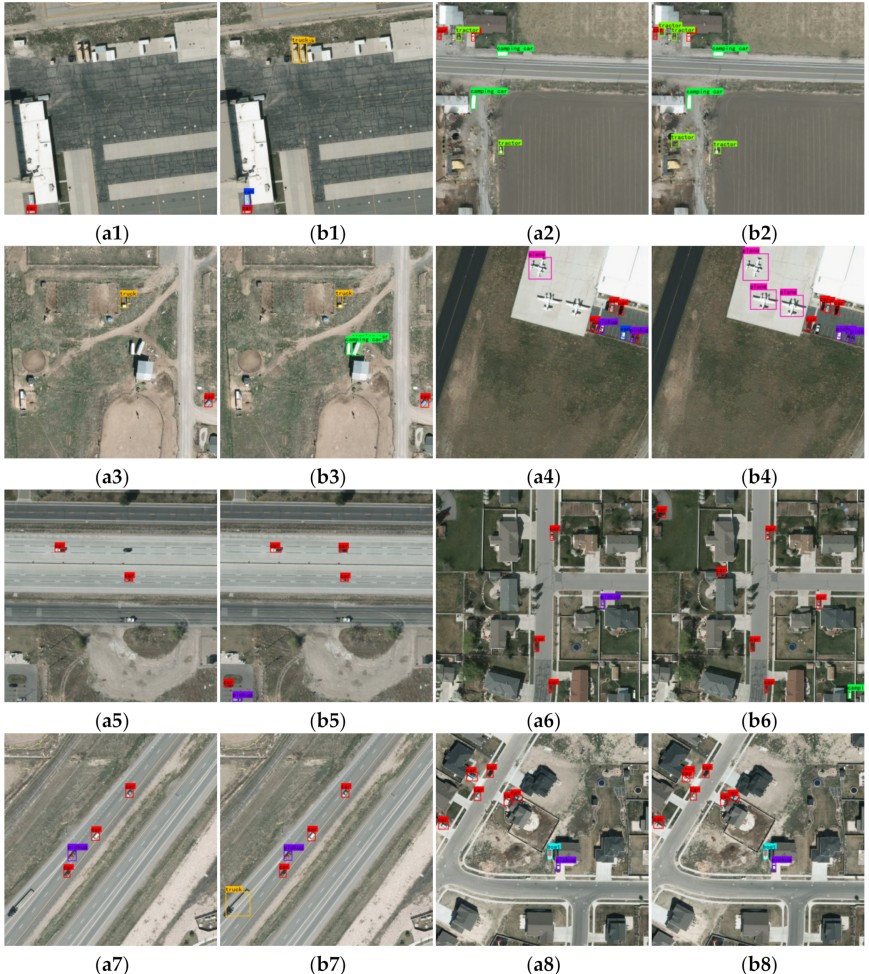

**Figure 9.** Detection results on the VEDAI dataset: (**a1**–**a8**) and (**b1**–**b8**) are the detection results of YOLOv4 and our proposed method, respectively.

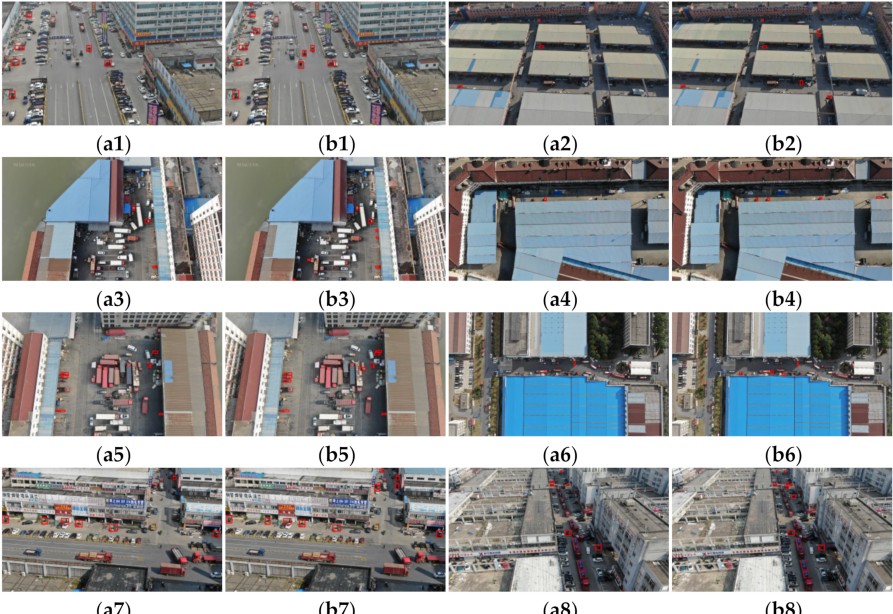

**Figure 10.** Detection results on the forklift dataset: (**a1**–**a8**) and (**b1**–**b8**) are the detection results of YOLOv4 and our proposed method, respectively.

## 4. Conclusions

In this paper, we analyzed the shortcomings of current detection methods according to the characteristics of UAV aerial images. We proposed a new method for UAV aerial images, which were improved on the basis of YOLOv4. In the backbone, we added a new attention module, the IECA module. This module efficiently utilized the interrelationships between channels, which was helpful for model feature extraction. According to the characteristics of different activation functions, we used HardSwish and Mish activation functions in the shallow and deep layers of the network, respectively. This reduced model complexity while also providing a good detection effect. When calculating the bounding box regression loss, using the EIOU loss function speeds up the convergence and improves the localization effect. At the neck of the model, the SPP module was replaced with a pyramid pooling module, and an ASFF feature fusion module was added at the end. Such improvements help to expand the receptive field of the model and strengthen feature fusion. We established a forklift dataset based on drone aerial imagery, which consisted of 1007 annotated images. We conducted a series of experiments on the PASCAL VOC, VEDAI, and forklift datasets. By conducting ablation experiments, the effectiveness of the proposed attention module and the rest of the improvements were verified. Among them, the improvements of the activation functions and loss function were more universal. A well-designed activation function is conducive to improving the propagation of gradients and reducing the computational cost of the model, and a well-designed loss function is conducive to measuring the consistency of the prediction results with the real situation, making the target regression more stable. Through comparative experiments, the superiority of the proposed model among several other advanced detection methods was verified. Finally, the experimental results for UAV aerial images demonstrated that the remaining improvements are more related to the characteristics of UAV aerial images, which are suitable for images containing rich, small targets and complex backgrounds. Compared with YOLOv4, this method has good detection performance.

We will continue to research target detection methods for UAV aerial images in the future. There is a need to better balance model complexity and detection accuracy, and to validate the model with integrated data from several sources [59]. In this paper, although the computational cost of the model was reduced in terms of the choice of activation function, the rest of the improvements will lead to an increase in computational cost due to the addition of modules. We will tweak the network structure even further to boost the model's detection performance in UAV aerial images.

**Author Contributions:** Conceptualization, X.L. and Y.W.; Data curation, X.L. and L.Z.; Formal analysis, X.L.; Funding acquisition, Y.W.; Investigation, X.L.; Methodology, X.L.; Project administration, X.L. and Y.W.; Software, X.L.; Validation, X.L.; Visualization, X.L.; Writing—original draft, X.L. and Y.W.; Writing—review and editing, X.L. All authors have read and agreed to the published version of the manuscript.

**Funding:** This research was funded by the National Nature Science Founding of China grant number 61573183.

**Institutional Review Board Statement:** Not applicable.

**Informed Consent Statement:** Not applicable.

**Acknowledgments:** This research was funded by Wuxi Gewu Intelligent Technology Co., Ltd. Thanks to the equipment and personnel support provided by Wuxi Gewu Intelligent Technology Co., Ltd.

**Conflicts of Interest:** The authors declare no conflict of interest.

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
