# Peer review of "YOLOD: A Target Detection Method for UAV Aerial Imagery"

_remotesensing, doi:10.3390/rs14143240_

Round 1
Reviewer 1 Report
This paper achieved good results on the PASCAL VOC, VEDAI dataset by improving the YOLOv4 model. I have the following questions about the experiments,
1. Why do you choose the optimizer as SGD instead of Adam or other good optimizers? As far as I know SGD usually doesn't get the best results.
2. For the ablation experiment, I think two groups need to be added: 1) Only change the Activation Function. 2) Only change the Loss Function.
This shows whether both improvements are valid and what improvements are more important to the model.
3. Why do you choose YOLOv4 instead of YOLOv5 or YOLOX which is better for improvement?
4. I don’t think that the model control group is comparative using Faster RCNN and SDD. Their backbone network is outdated, and the results also show that the model is not as effective as the original YOLOv4. You should use more advanced models such as YOLOv5 or more advanced backbone networks.
Reviewer 2 Report
Dear Authors, Your article is very interesting. In the attachment, I send you my suggestion

Reviewer 3 Report
This paper proposed an improved YOLOv4 method using activation functions suitable for UAV aerial images. The technique is well described and interesting results are demonstrated. However, there are some points improvement and clarification are needed as below.
It is mentioned that the proposed method is suitable for UAV images, but the improvement in VOC2007 results is better in mAP. Does the proposed method have universality?
Does it mean that methods such as the adjusted activation function introduced according to the characteristics of UAV images are much more universal?
What is the difference in computational cost from existing methods?
Round 2
Reviewer 3 Report
Dear authors,
I appreciated the improvement done on this revision. The clarity of the paper is improved. The manuscript can be published as it is.